

# Effects of irrigation and nitrogen management on phyllosphere microbial communities of silage maize

Liuxing Xu[1,2], Changjing Chen[2], Chenggang He[1], Ahmed M. Abd El Tawab[3], Qinhua Liu[1] and Hua Jiang[1]

[1] College of Animal Science and Technology, Yunnan Agriculture University, Kunming, China
[2] College of Agronomy and Life Sciences, Zhaotong University, Zhaotong, China
[3] Dairy Science Department, National Research Centre, Giza, Egypt

## ABSTRACT

Silage maize (*Zea mays*) is a significant source of animal roughage in many countries. Few studies have revealed the specific impacts of agronomic measures on harmful or beneficial microbial species (based on plant health or utilization) in silage maize. The aim of the present study was to investigate the effects of three maturity stages (big trumpet, milk, and dough) × two irrigation amounts (1,200 $m^3$ $hm^{-2}$ (IA1,200) and 2,400 $m^3$ $hm^{-2}$ (IA2,400)) × three nitrogen (N) application rates (160 kg $hm^{-2}$ (low), 240 kg $hm^{-2}$ (medium), and 320 kg $hm^{-2}$ (high)) on the bacterial community structure of the silage maize phyllosphere. Irrigation amounts and N application rates did not affect silage maize leaves' chemical or physiological properties, but influenced the bacterial community structure of silage maize phyllosphere. There were higher relative abundances of beneficial (*Pseudomonas*, *Rhodococcus*, *Achromobacter*, and *Myroides*) or harmful (*Bordetella* and *Ralstonia*) microbial in IA2,400 compared with IA1,200. Plant pathogenic bacteria (*Erwinia* and *Serratia*) were found to have the highest relative abundance at the low N application rates. In meta-analysis, some bacteria impacted the phytosanitation and nutrition quality of forage, encompassing *Ralstonia*, *Pantoea*, *Dokdonella*, *Vogesella*, *Erwinia*, *Serratia*, *Pseudomonas*, and *Bordetella*. Based on the yield, plant health, and potential fermentation quality of silage maize, we recommend using an irrigation amount of 2,400 $m^3$ $hm^{-2}$ and an N application rates of 240 kg $hm^{-2}$ for agricultural production and harvesting silage maize at the milk stage.

## INTRODUCTION

Silage maize (*Zea mays*) has become a significant source of roughage in many countries considering the yield and animal digestion rate (*Li et al., 2010*). Globally, over 133 million dairy cows consume approximately $665 \times 10^8$ tons of silage annually, with whole-plant maize silage accounting for over 40% of the silage consumed (*Xu et al., 2021*). Countries such as Canada have specifically bred silage maize with shorter growing seasons to address the issue of insufficient warmth (*Guyader, Baron & Beauchemin, 2018*). In the United States, over $2.4 \times 10^5$ $hm^2$ of silage maize is planted to provide sufficient roughage

Corresponding authors
Qinhua Liu, liuqinhua@njau.edu.cn
Hua Jiang, jianghua15@163.com

(*Bernard & Tao, 2020*). In China, large quantities of silage maize are planted in regions such as Heilongjiang, Shaanxi, and Inner Mongolia to produce silage for animal feed or sale (*Wang et al., 2021*). In Brazil, silage maize is planted in the spaces among trees to obtain more roughage (*Teodoro et al., 2023*). Despite its widespread use, challenges remain regarding the improvement of its nutritional and hygienic quality.

Phyllosphere bacteria, located within and on the surface of leaves, are closely related to phytosanitation. For example, *Myroides* (*Kaur & Kaur, 2021*) and *Rhodococcus* (*Afordoanyi et al., 2023*) are popular for their ability to improve crop root growth and degrade mycotoxins, respectively, while *Ralstonia* is often noted for its positive correlation with plant morbidity (*Ahmed et al., 2022*). Additionally, phyllosphere bacteria play important roles in the quality and safety of fodder, such as nutrition and hygienic quality of silage. Specifically, phyllosphere *Lactiplantibacillus*, *Weissella*, and *Pediococcus* are popular for improving the fermentation quality of silage (*Zhou et al., 2024*). However, phyllosphere *Enterobacter* and *Clostridium* are undesired genera as they cause significant protein degradation, in addition to spore and toxin degradation during ensiling (*Wang et al., 2019*; *Zhang et al., 2019b*). Furthermore, *Acetobacter* oxidize lactic acid and acetic acid, leading to increase in pH and carbon dioxide content; thereby causing a loss of dry matter (*Avila & Carvalho, 2020*). Such undesired bacteria preliminarily influenced the structure of microbial communities at the initial ensiling stage, followed by defining the final nutrition and hygienic quality of silage. Therefore, controlling phyllosphere undesired bacteria is key to phytosanitation, nutrition and hygienic quality of silage. However, the species, structure, and physiological and biochemical properties of plants related to epiphytic microorganismsare not well understood (*Tang et al., 2023*), and little information is available about the factors that alter the survival and distribution of undesired phyllosphere bacteria. Moreover, the current understanding of how production management processes influence the mechanisms of these bacteria remains limited, and the specific regulatory pathways and functional mechanisms have yet to be systematically elucidated.

Agronomic measures, such as fertilization and irrigation, are common approaches to managing plant production. Nitrogen (N) is a key nutrient in plant production, and its supply directly affects yield and quality. However, naturally occurring N in the soil often fails to meet the demands of plant growth. Therefore, artificially N fertilizer has become the main N source in crop production (*Canfield, Glazer & Falkowski, 2010*). Some studies have observed changes in the species of bacteria attached to forage and their relative abundance after N fertilizer application; such bacteria were closely related to plant phytosanitation and nutrition, in addition to thehygienic quality of silage (*Chen, Dong & Zhang, 2021*; *Wu et al., 2023a*). However, the underlying reasons for these changes are not known.

In most countries, rainfall is the primary source of water for plant growth; however, uneven temporal and spatial distributions of rainfall often lead to drought or waterlogging damage to forage (*Kim & Sung, 2023*; *Walter et al., 2012*). Under drought stress, plants adjust their nutritional sources by regulating soluble carbohydrates, total phenolic compounds, and free amino acid content (*Habus Jercic et al., 2023*); ultimately determining bacterial population size and structure (*Yadav, Karamanoli & Vokou, 2005*). In addition, drought decreases the diversity of epiphytic bacterial communities in grasses, with some

bacteria increasing their drought resistance by thickening their cell walls (*Bechtold et al., 2021*). However, these phyllosphere bacteria were caused by irrigation alteration whether change in the plant phytosanitation, and nutrition and hygienic quality of silage was little known.

The objective of this study aimed to address the existing knowledge gap that how N application and irrigation directly or indirectly influence the bacterial communities in the phyllosphere of silage maize. Specifically, we hypothesizes that agronomic practices such as fertilization and irrigation influence the composition and function of phyllosphere bacterial communities, which in turn affect silage fermentation processes, ultimately shaping the nutritional value and hygienic quality of the final product. To gain insights into the underlying microbial dynamics, single-molecule real-time (SMRT) sequencing technology was employed. The findings are expected to provide valuable predictive information for improving crop safety and silage nutrient content, ultimately contributing to the production of hygienic and high-quality silage.

## MATERIALS AND METHODS

### Experimental treatment and crop management

The field experiment was conducted from Apr. 2022 to Nov. 2022 at the Experimental Field of Zhaotong University (27°36′65′N, 103°74′63′E, altitude 1,912 m; Guoxue Road, Zhaoyang district, Yunnan province, China). The experiment was designed with randomized block experiment included three factors: three maturity stages (big trumpet, milk, and dough) × two irrigation amounts (IA, 1,200 $m^3$ $hm^{-2}$ (IA1,200) and 2,400 $m^3$ $hm^{-2}$ (IA2,400)) × three N application rates (NARs, 160 kg $hm^{-2}$ (low), 240 kg $hm^{-2}$ (medium), and 320 kg $hm^{-2}$ (high)) × three replicates (including four healthy plants). To minimize the environmental impact on phyllosphere bacteria, the silage maize was planted in foam boxes (inner diameter: 540 mm, width: 385 mm, height: 300 mm). A total of 54 foam boxes were used. The soil required for the experiment was evenly mixed, and 13 kg of soil (lithic phaeozem, natural moisture content of 23%) was filled into each foam box, with a spacing of 50 cm between the boxes. Eight seeds were evenly sown in each box on Apr. 27, 2022. After emergence, seedlings with significant growth differences or unhealthy growth were removed to ensure that each box had two evenly growing and distributed healthy seedlings (Planting density: 100,000 plant $hm^{-2}$). Silage maize was harvested on Nov. 13, 2022. During culture, N fertilizer was applied at a rate of 40% at the seedling stage (May 10, 2022) and 60% at the jointing stage (July 3, 2022). Nitrogen fertilizer, containing 46% nitrogen in the form of urea, was evenly broadcast directly onto the soil. At the vegetative growth stage, water was applied every seven days. At the reproductive growth stage, water was applied every five days. When watering, the soil moisture content was maintained at 65%–80% (10 cm from the soil surface) and determined using a microwave oven. Irrigation was performed manually, ensuring that water was evenly distributed across the soil surface during the process. The irrigation volumes during the vegetative growth and reproductive growth stages accounted for 45% and 55% of the total irrigation volume, respectively, with 14 and 18 irrigation events conducted for each stage. Specifically, 15% of the total irrigation

volume was applied before the jointing stage, 30% was applied from the jointing to the big trumpet stage, and 55% was applied after the big trumpet stage. The foam boxes containing the planted silage maize were placed in a greenhouse with a transmittance rate of 85%. No pesticides were used during the growth of silage maize due to using the anti-bug net.

## Field sampling

Leaves were collected at the big trumpet (August 13, 2022; Bologische Bundesanstalt, Bundessortenamt and Chemical Industry scale (BBCH): 55), milk (September 17, 2022; BBCH: 75), and dough (November 13, 2022; BBCH: 87) stages. The sampling was conducted on different individual plants each time to avoid cumulative damage. On sunny mornings, mature and healthy leaves from the same location were selected (leaf surface was free of dew), avoiding veins and leaf edges. The photosynthetic properties were measured using an LI-6800 portable photosynthesis system (LI-COR Biosciences, Lincoln, NE, USA). The maize leaves were collected and divided into three parts (sterile conditions). The first part was immediately transported using ice packs to (stored at 4 °C) to measure microbial numbers and chemical properties. The second part was placed on ice packs for subsequent analysis of the structural and physiological properties. The third part was stored in liquid N for subsequent determination of the bacterial species and relative abundance.

## Physiological properties of leaf

The tape measure was placed at the base of the leaf where it connected to the plant stem. The leaf was gently straightened, and the length to the leaf tip was measured along the midrib, ensuring that the tape measure made contact with the leaf surface and remained horizontal for accurate measurement. The width at the widest point of the leaf was measured. Typically, this position was in the middle of the leaf or slightly towards the tip. Similar care was taken with the tape measure to ensure accurate measurement. The total chlorophyll content was calculated based on the content of chlorophyll a and chlorophyll b (*Porra, Thompson & Kriedemann, 1989*). The leaf moisture retention capacity (MRC) was calculated using the fresh weight of the leaf, weight after 5 h of natural dehydration, and weight after drying at 75 °C until a constant weight was determined (*Ni et al., 2015*). The photosynthetic properties, such as transpiration rate (Tr), net photosynthetic rate (Pn), intercellular carbon dioxide concentration (Ci), pore conductivity of water vapor (Gsw), total conductivity of water vapor (Gtw), and total conductivity of carbon dioxide (Gtc), were analyzed using a LI-6800, and the parameter settings are consistent with those reported earlier (*Chen, Dong & Zhang, 2021*; *Wu et al., 2023a*).

## Chemical properties of plant

The leaves were snap frozen in liquid N and ground into powder. The soluble protein content of the leaves was determined using the Coomassie brilliant blue method (*Bradford, 1976*). Total phenolic content was measured using a UV-visible spectrophotometer (*Yoo et al., 2004*). Water-soluble carbohydrate content was measured using a sucrose-based standard curve and the anthrone-sulfuric acid method (*Murphy, 1958*). The phosphate content was determined using the phosphomolybdate blue colorimetric method (*Hande*

*et al., 2013*). The free amino acid content was determined using ninhydrin as a coloring agent (*Lee & Takahashi, 1966*).

## Microbial population

Fresh material (10 g) was placed in a sterile polyethylene bag, 90 mL of sterile distilled water was added, and the bag was placed on a shaker for 10 min to obtain suspensions for microbial determination (*Chen, Dong & Zhang, 2021*). Potato dextrose, nutrient, and De Man–Rogosa–Sharpe agar media were used for the aerobic culture of yeast and molds at 37 °C for 48 h, aerobic culture of aerobic bacteria at 37 °C for 24 h, and anaerobic culture of lactic acid bacteria at 37 °C for 48 h, respectively. The results were converted to lg cfu $g^{-1}$ FW (cfu stands for colony-forming units, FW stands for fresh weight).

## Analysis of bacterial community

Total DNA was extracted from the bacteria using the E.Z.N.A. Plant DNA Kit (Omega Bio-tek, Norcross, GA, USA). The quality of the extracted DNA was assessed using 1% agarose gel electrophoresis. The DNA concentration and purity were determined using a NanoDrop™ 2000 (Thermo Fisher Scientific, Waltham, MA, USA). Polymerase chain reaction (PCR) amplification of the V3–V4 region of the 16S rRNA gene was performed using primers 779F (5′-ACTCCTACGGGAGGCAGCAG-3′) and 1193R (5′-GGACTACHVGGGTWTCTAAT-3′). After mixing the PCR products from the same sample, PCR product recovery was performed on a 2% agarose gel. Gel electrophoresis with 2% agarose was conducted for detection, using AxyPrepDNA Gel Recovery Kit (Axygen Scientific Inc., Union City, CA, USA) for excising PCR products from the gel, followed by Tris_HCl elution. A Quantus™ Fluorometer (Promega, Madison, WI, USA) was used to detect and quantify the recovered products. Library construction of the purified PCR products was performed using the NEXTFLEX Rapid DNA-Seq Kit, followed by sequencing on an Illumina PE300 platform (San Diego, CA, USA). The paired-end reads were first assembled based on overlapping relationships, while simultaneously undergoing quality control and sequence quality filtering. Following sample differentiation, operational taxonomic unit (OTU) clustering analysis and taxonomic classification were performed. Diversity index analyses were conducted using OTUs. The results of the OTU clustering analysis were used for multiple diversity index analyses, along with an assessment of sequencing depth (*Tkacz et al., 2020*).

## Meta analysis data retrieval and process

A comprehensive literature search was conducted using the following search engines: Web of Science (deadline: Aug. 05 2024). Publications were retrieved using the term: phyllosphere microbial (1,723 publications) and leaf microbial (30,710 publications). Of the articles that were retrieved, only those that satisfied the predetermined inclusion criteria were included in the meta-analysis. For inclusion into the meta-analysis, studies needed to have the following (standardized criteria): (1) the control treatment does include any forage (60 publications) or grass (61 publications) (plant health dataset); (2) include treatments comprising only with yield (146 publications) (yield dataset); (3) include treatments comprising only with silage fermentation quality (14 publications)

(silage fermentation quality). A flowchart explaining the process of study identification and selection for analyzing the effects of phyllosphere microbial communities on plant health and silage fermentation quality were shown in Fig. S1. And calculate the influence of harmful bacteria on silage fermentation quality based on the obtained data.

### Statistical analysis

Statistical analyses were performed based on 54 biological replicates per treatment ($n = 54$), ensuring the reliability of the results. Analysis of variance (ANOVA) was applied to assess the significance of maturity stage, IA, and NAR on phyllosphere bacterial abundance and diversity, followed by Duncan's *post-hoc* test (IBM SPSS Statistics 26; IBM Corp., Armonk, NY, USA) to identify specific group differences. Pearson's correlation analysis was conducted to reveal potential linkages between environmental factors and microbial parameters. The normalized shuffle test quantified community stability under different treatments, where higher values (>50%) indicate stochastic assembly dominance (*Li et al., 2018*). All models incorporated treatment effects and residuals, with assumptions (*e.g.*, normality, homogeneity). Furthermore, we employed principal component analysis (PCA) to: (a) reduce the dimensionality of the multivariate dataset while preserving maximum variance, (b) visualize the complex relationships between phyllosphere microbial communities and environmental factors, and (c) identify potential patterns/clusters that might reflect underlying biological mechanisms. This study employed an environment factor-based bacterial community assessment approach to systematically analyze the regulatory effects of environmental factors on bacterial community structure by quantifying the associations between key environmental parameters and microbial abundance/diversity indices. The analytical method has been previously applied to decipher the explanatory power of environmental factors on bacterial community variation (*Wu et al., 2023a*; *Zhang et al., 2022*). Compared with simple correlation analysis, this method enables simultaneous evaluation of synergistic effects among multiple environmental factors, with statistical significance assessed through 999 permutation tests, thereby providing more comprehensive insights into environment-microbiome interactions.

## RESULTS

### The highest nutrient value was at the milk stage

As shown in Table 1, soluble protein content was influenced by maturity stage and maturity stage $\times$ IA. The contents of total phenol, water-soluble carbohydrates, phosphorus, and free amino acid were only affected by the maturity stage ($P < 0.05$). The contents of soluble protein, phosphorus, and free amino acid were higher ($P < 0.05$) at the big trumpet stage by 4.97 µg g$^{-1}$, 67 µg kg$^{-1}$, and 4.80 mg g$^{-1}$ and were higher ($P < 0.05$) at the milk stage by 4.27 µg g$^{-1}$, 147 µg kg$^{-1}$, and 8.05 mg g$^{-1}$ relative at the dough stage.

### Moderate NAR and IA2,400 increased nutrient value

As irrigation increased from IA1,200 to IA2,400, the soluble protein (5.04 µg g$^{-1}$ *vs.* 6.28 µg g$^{-1}$), total phenol (8.81 g kg$^{-1}$ *vs.* 9.59 g kg$^{-1}$), water-soluble carbohydrate (15.5 mg g$^{-1}$ *vs.* 18.3 mg g$^{-1}$), and phosphorus (228 µg g$^{-1}$ *vs,* 251 µg g$^{-1}$) contents slightly increased

**Table 1  Chemical properties, leaf length, and leaf width of different maturity stage, irrigation amount, and N application rate ($n = 54$).**

| Maturity stage and treatment | | SP ($\mu$g g$^{-1}$) | TP (g kg$^{-1}$) | WSC (mg g$^{-1}$) | P ($\mu$g g$^{-1}$) | FAA (mg g$^{-1}$) | LL (cm) | LW (cm) |
|---|---|---|---|---|---|---|---|---|
| Maturity stage (MS) | Big trumpet stage | $7.55 \pm 0.13$a | $8.88 \pm 0.65$b | $2.27 \pm 0.32$c | $235 \pm 16.8$b | $6.27 \pm 0.51$b | $105 \pm 2.16$ | $11.2 \pm 0.31$ |
| | Milk stage | $6.85 \pm 0.57$a | $4.10 \pm 0.19$c | $7.83 \pm 0.82$b | $315 \pm 20.0$a | $9.52 \pm 0.87$a | $98.3 \pm 2.56$ | $11.3 \pm 0.37$ |
| | Dough stage | $2.58 \pm 0.07$b | $14.6 \pm 0.59$a | $40.6 \pm 1.91$a | $168 \pm 14.1$c | $1.47 \pm 0.15$c | $96.9 \pm 2.96$ | $11.4 \pm 0.29$ |
| Irrigation amount (IA) | IA1200 | $5.04 \pm 0.42$ | $8.81 \pm 0.90$ | $15.5 \pm 3.40$ | $228 \pm 17.5$ | $5.90 \pm 0.88$ | $99.3 \pm 2.49$ | $10.5 \pm 0.24$b |
| | IA2400 | $6.28 \pm 0.56$ | $9.59 \pm 0.97$ | $18.3 \pm 3.50$ | $251 \pm 18.5$ | $5.60 \pm 0.71$ | $101 \pm 1.86$ | $12.1 \pm 0.19$a |
| N application rate (NAR) | Low | $5.44 \pm 0.65$ | $8.72 \pm 0.96$ | $16.7 \pm 4.27$ | $239 \pm 25.0$ | $5.88 \pm 1.16$ | $99.8 \pm 2.52$ | $11.3 \pm 0.27$ |
| | Medium | $5.80 \pm 0.59$ | $9.67 \pm 1.35$ | $16.8 \pm 4.05$ | $262 \pm 23.4$ | $5.63 \pm 0.81$ | $99.5 \pm 2.44$ | $11.5 \pm 0.25$ |
| | High | $5.74 \pm 0.65$ | $9.21 \pm 1.14$ | $17.3 \pm 4.50$ | $219 \pm 16.6$ | $5.74 \pm 0.98$ | $101 \pm 3.14$ | $11.1 \pm 0.41$ |
| P value | MS | *** | *** | *** | *** | *** | NS | NS |
| | IA | NS | NS | NS | NS | NS | NS | *** |
| | NAR | NS | NS | NS | NS | NS | NS | NS |
| | MS×LA | *** | NS | NS | NS | NS | NS | NS |
| | MS×NAR | NS | NS | NS | NS | NS | NS | NS |
| | IA×NAR | NS | NS | NS | NS | NS | NS | NS |
| | MS×LA×NAR | NS | NS | NS | NS | NS | NS | NS |
| Standard error of the means | | 0.36 | 0.66 | 2.42 | 12.7 | 0.56 | 0.09 | 0.18 |

**Notes.**

The values represent the mean $\pm$ standard error . Different lowercase letters in the same column represent significant difference among maturity stage, irrigation amount or N application rate ($P < 0.05$); the absence of lowercase letters indicated that there were no significant differences among maturity stage, irrigation amount or N application rate ($P > 0.05$).

*** significant at $P < 0.001$.

NS, not significant; SP, soluble protein; TP, total phenol; WSC, water-soluble carbohydrates; P, phosphorus; FAA, free amino acid; LL, leaf length; LW, leaf width; IA1200, irrigation amount was 1,200 m$^3$ hm$^{-2}$; IA2400, irrigation amount was 2,400 m$^3$ hm$^{-2}$; Low, N application rate was 200 kg hm$^{-2}$; Medium, N application rate was 350 kg hm$^{-2}$; High, N application rate was 500 kg hm$^{-2}$.

($P > 0.05$); whereas the content of free amino acid (5.90 mg g$^{-1}$ *vs.* 5.60 mg g$^{-1}$) slightly decreased ($P > 0.05$). Higher ($P < 0.05$) leaf width was in the IA2,400 compared with IA1,200 (Table 1).

## The highest physiological properties were at the milk stage

The physiological properties, such as chlorophyll content, MRC, Tr, Pn, Ci, Gsw, Gtw, and Gtc, varied with maturity (Table 2) ($P < 0.001$). Chlorophyll contents at the big trumpet and milk stages were higher ($P < 0.05$) than at the dough stage. The MRC of leaves at the milk stage decreased ($P < 0.05$) by about 12.6% and 15.7% relative at the big trumpet and dough stages. Additionally, the Tr, Pn, Gsw, Gtw, and Gtc to the milk stage were significantly higher ($P < 0.05$) than those at the big trumpet and dough stages. As irrigation increased from IA1,200 to IA2,400, the chlorophyll, Tr, Pn, Gsw, Gtw, and Gtc in the leaves slightly increased ($P > 0.05$); whereas the MRC (37.1% *vs.* 36.4%) and Ci (174 $\mu$mol mol$^{-1}$ *vs.* 173 $\mu$mol mol$^{-1}$) slightly decreased ($P > 0.05$). From low to high NAR, the Tr, Pn, Ci, Gsw, Gtw, and Gtc in silage maize slightly increased ($P > 0.05$).

## Maturity stage and IA affected the numbers of aerobic bacteria and molds

Different factors affected the microbial numbers. Aerobic bacteria were affected by maturity stage and IA ($P < 0.05$); lactic acid bacteria was affected by maturity stage × IA and IA ×

**Table 2  Physiological properties of different maturity stage, irrigation amount, and N application rate ($n = 54$).**

| Maturity stage and treatment | | Chlorophyll (mg g$^{-1}$) | Moisture retention capacity (%) | Transpiration rate (mmol m$^{-2}$ s$^{-1}$) | Net photosynthetic rate (μmol m$^{-2}$ s$^{-1}$) | Intercellular carbon dioxide concentration (μmol mol$^{-1}$) | Pore conductivity of water vapor (mmol m$^{-2}$ s$^{-1}$) | Total conductivity of water vapor (mmol m$^{-2}$ s$^{-1}$) | Total conductivity of carbon dioxide (mmol m$^{-2}$ s$^{-1}$) |
|---|---|---|---|---|---|---|---|---|---|
| Maturity stage (MS) | Big trumpet stage | 2.40 ± 0.14a | 39.9 ± 1.27a | 0.81 ± 0.08b | 2.95 ± 0.40c | 302 ± 20.2a | 43.4 ± 4.71b | 42.9 ± 4.60b | 27.4 ± 2.82b |
| | Milk stage | 2.49 ± 0.11a | 27.3 ± 1.54b | 3.46 ± 0.23a | 31.2 ± 0.95a | 79.1 ± 3.84c | 162 ± 10.9a | 157 ± 10.4a | 98.5 ± 6.54a |
| | Dough stage | 1.69 ± 0.13b | 43.0 ± 2.80a | 1.22 ± 0.08b | 8.61 ± 0.93b | 140 ± 15.9b | 57.3 ± 3.81b | 56.6 ± 3.72b | 35.4 ± 2.34b |
| Irrigation amount (IA) | IA1200 | 2.04 ± 0.13 | 37.1 ± 1.90 | 1.59 ± 0.24 | 13.9 ± 2.45 | 174 ± 26.1 | 76.0 ± 11.5 | 74.2 ± 11.0 | 46.9 ± 6.85 |
| | IA2400 | 2.34 ± 0.11 | 36.4 ± 2.25 | 2.07 ± 0.27 | 14.9 ± 2.51 | 173 ± 17.1 | 99.4 ± 12.0 | 96.6 ± 11.4 | 60.6 ± 7.19 |
| N application rate (NAR) | Low | 2.18 ± 0.16 | 37.1 ± 2.38 | 1.62 ± 0.26 | 13.4 ± 2.55 | 164 ± 21.7 | 77.8 ± 12.0 | 76.1 ± 11.5 | 48.2 ± 7.12 |
| | Medium | 2.08±0.16 | 36.1 ± 2.57 | 1.77 ± 0.33 | 13.8 ± 3.28 | 173 ± 25.6 | 85.0 ± 15.2 | 82.7 ± 14.6 | 51.9 ± 9.17 |
| | High | 2.31 ± 0.13 | 37.0 ± 2.77 | 2.09 ± 0.35 | 15.6 ± 3.30 | 184 ± 33.1 | 100 ± 16.2 | 97.4 ± 15.4 | 61.1 ± 9.72 |
| *P* value | MS | *** | *** | *** | *** | *** | *** | *** | *** |
| | IA | NS | NS | NS | NS | NS | NS | NS | NS |
| | NAR | NS | NS | NS | NS | NS | NS | NS | NS |
| | MS×LA | NS | NS | NS | NS | ** | NS | NS | NS |
| | MS×NAR | NS | NS | ** | ** | NS | ** | ** | ** |
| | IA×NAR | NS | NS | NS | NS | NS | NS | NS | NS |
| | MS×LA×NAR | NS | NS | ** | NS | NS | ** | ** | ** |
| Standard error of the means | | 0.11 | 1.46 | 0.18 | 1.74 | 15.4 | 8.37 | 7.99 | 5.01 |

**Notes.**

The values represent the mean ± standard error. Different lowercase letters in the same column represent significant difference among maturity stage, irrigation amount or N application rate ($P < 0.05$); the absence of lowercase letters indicated that there were no significant differences among maturity stage, irrigation amount or N application rate ($P > 0.05$).

*** significant at $P < 0.001$.

** significant at $P < 0.01$.

NS, not significant; IA1200, irrigation amount was 1,200 m$^3$ hm$^{-2}$; IA2400, irrigation amount was 2,400 m$^3$ hm$^{-2}$; Low, N application rate was 200 kg hm$^{-2}$; Medium, N application rate was 350 kg hm$^{-2}$; High, N application rate was 500 kg hm$^{-2}$.

**Table 3  Microorganism compositions of different maturity stage, irrigation amount, and N application rate ($n = 54$).**

| Maturity stage and treatment | | Aerobic bacteria (lg cfu g$^{-1}$ FM) | Lactic acid bacteria (lg cfu g$^{-1}$ FM) | Yeast (lg cfu g$^{-1}$ FM) | Molds (lg cfu g$^{-1}$ FM) |
|---|---|---|---|---|---|
| Maturity stage (MS) | Big trumpet stage | 5.19 ± 0.14b | 2.55 ± 0.13 | 4.49 ± 0.25 | 2.40 ± 0.10b |
| | Milk stage | 7.07 ± 0.32a | 2.65 ± 0.10 | 4.32 ± 0.11 | 2.67 ± 0.14ab |
| | Dough stage | 4.86 ± 0.15b | 2.49 ± 0.10 | 4.28 ± 0.12 | 2.87 ± 0.10a |
| Irrigation amount (IA) | IA1200 | 6.18 ± 0.25a | 2.60 ± 0.08 | 4.62 ± 0.13a | 2.83 ± 0.07a |
| | IA2400 | 5.22 ± 0.16b | 2.53 ± 0.09 | 4.10 ± 0.13b | 2.46 ± 0.11b |
| N application rate (NAR) | Low | 6.18 ± 0.37 | 2.49 ± 0.11 | 4.60 ± 0.13 | 2.62 ± 0.12 |
| | Medium | 5.73 ± 0.15 | 2.68 ± 0.09 | 4.11 ± 0.17 | 2.71 ± 0.14 |
| | High | 5.19 ± 0.22 | 2.53 ± 0.12 | 4.38 ± 0.19 | 2.61 ± 0.11 |
| P value | MS | *** | NS | NS | * |
| | IA | * | NS | ** | ** |
| | NAR | NS | NS | NS | NS |
| | MS×LA | NS | * | NS | NS |
| | MS×NAR | NS | NS | *** | NS |
| | IA×NAR | NS | ** | NS | NS |
| | MS×LA×NAR | NS | NS | ** | NS |
| Standard error of the means | | 0.23 | 0.06 | 0.10 | 0.07 |

Notes.

The values represent the mean ± standard error. Different lowercase letters in the same column represent significant difference among maturity stage, irrigation amount or N application rate ($P < 0.05$); the absence of lowercase letters indicated that there were no significant differences among maturity stage, irrigation amount or N application rate ($P > 0.05$).

*** significant at $P < 0.001$.

** significant at $P < 0.01$.

* significant at $P < 0.05$.

NS, not significant; IA1200, irrigation amount was 1,200 m$^3$ hm$^{-2}$; IA2400, irrigation amount was 2,400 m$^3$ hm$^{-2}$; Low, N application rate was 200 kg hm$^{-2}$; Medium, N application rate was 350 kg hm$^{-2}$; High, N application rate was 500 kg hm$^{-2}$.

NAR ($P < 0.05$); yeast was affected by IA, maturity stage × NAR, and maturity stage × IA × NAR ($P < 0.05$); and molds was affected by maturity stage and IA ($P < 0.05$) (Table 3). The highest numbers of aerobic bacteria and molds were observed at the milk and dough stages, respectively ($P < 0.05$). At IA1,200, the aerobic bacteria, yeast, and molds were increased ($P < 0.05$) by 0.96 lg cfu g$^{-1}$ FM, 0.52 lg cfu g$^{-1}$ FM, and 0.37 lg cfu g$^{-1}$ FM, respectively, compared with IA2,400. With increasing NAR, the number of aerobic bacteria slightly decreased ($P > 0.05$). The highest numbers of lactic acid bacteria and molds were observed at the medium NAR.

## Aerobic bacteria numbers were easily affected by environmental factors

Correlation analyses showed a significant positive correlation between the numbers of aerobic bacteria and free amino acid content, Tr, Pn, Gsw, Gtw, and Gtc ($P < 0.05$) (Fig. 1A). The soluble protein content was negatively correlated with yeast numbers ($P < 0.05$); the total phenol, water-soluble carbohydrate contents, MRC, and Ci were negatively correlated with aerobic bacteria numbers ($P < 0.05$). Meanwhile, free amino acid content, Tr, Pn, Gsw, Gtw, and Gtc were positively correlated with aerobic bacteria

numbers ($P < 0.05$). The first two components in the PCA explained 49.4% of the total variance and there was no overlap in the 95% confidence intervals for the three maturity stages (Fig. 1B, $P < 0.05$). The overlap in the 95% confidence intervals for the two irrigation levels was high at all three maturity stages (Figs. 1C–1E, $P > 0.05$). Similar results were obtained from different irrigation levels, indicating that, regardless of the maturity stage or irrigation level, there was a high overlap in the 95% confidence intervals for the three NAR (Fig. 2, $P > 0.05$).

## IA and NAR affect bacterial composition and diversity

For IA and NAR, the OTU ranges at the genus level, were 122–164 and 88–154, respectively (Figs. 3A and 3B). Among all the species, *Bordetella*, *Pantoea*, and *Pseudomonas* had the highest abundances (Figs. 3C and 3D). With increasing irrigation levels, the abundances of most species slightly decreased ($P > 0.05$) Notably, there was significantly lower ($P < 0.05$) abundance of *Lactobacillus* and *Rosenbergiella* in IA2,400 compared with IA1,200. Based on the 95% confidence intervals constructed for bacterial $\beta$ diversity, IA2,400 was almost entirely covered by IA1,200 (Fig. 3E). Similarly, high NAR were associated with low and medium NAR (Fig. 3F). The abundances of *Bordetella*, *Pseudomonas*, *Achromobacter*, *Rhodococcus*, *Ralstonia*, and *Myroides* in IA2,400 were lower than those in IA1,200 ($P < 0.05$) (Fig. 4A). The NAR only significantly affected ($P < 0.05$) the abundances of *Dokdonella*, *Roscomonas*, *Novosphingobium*, and *Vogesella* (Fig. 4B). For the NST index, IA1,200 was significantly higher than that IA2,400 ($P < 0.05$) , and the high NAR was significantly higher ($P < 0.05$) than the medium and low NARs (Figs. 4C and 4D).

## Environmental factors affected the abundance and species of bacteria

The IA1,200 and medium NAR were positively correlated with most environmental factors (Figs. 4E and 4F). Correlation analyses showed that there was a significant negative correlation ($P < 0.05$) between MRC and the relative abundances of *Achromobacter*, *Pseudomonas*, and *Delftia*; whereas MRC was positively correlated with *Pantoea* relative abundance ($P < 0.05$) (Fig. 4G). *Serratia* relative abundance was positively correlated with free amino acid content ($P < 0.05$). *Lactobacillus* relative abundance was negatively ($P < 0.05$) correlated with environmental factors, including free amino acid and chlorophyll contents, Tr, Gsw, Gtw, and Gtc. Among all environmental factors (Table 4), MRC shows the most significant impact on the changes of bacterial relative abundance ($R^2 = 0.2386$, $P = 0.0397$), while the impacts of the other environmental factors on bacterial relative abundance were relatively weak in comparison ($R^2 < 0.1$, $P > 0.05$).

## Meta-analysis of common harmful bacteria in fresh materials and natural silage

Based on meta analysis (Table 5), this study delved into the prevalent harmful bacteria found in fresh materials and natural silage without additives. These bacteria not only posed threats to human health through direct or indirect means but also manifested in the induction of diseases such as leaf spot, leading to a significant reduction in crop yield. Thereby exerting potential negative effects on agricultural production and silage

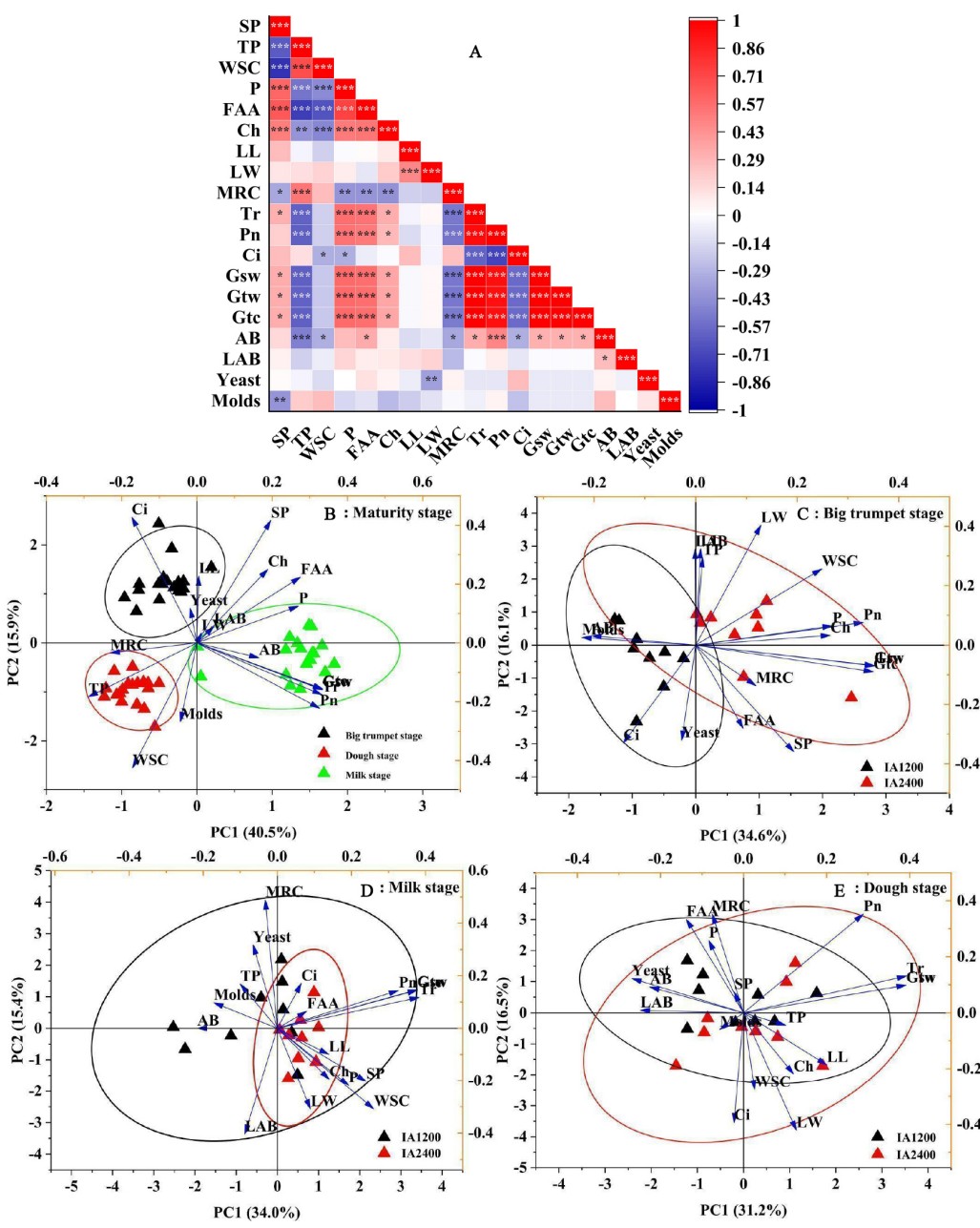

**Figure 1  Correlation plot of Pearson and principal component analyses of different maturity stage, irrigation amount.** AB, aerobic bacteria; LAB, lactic acid bacteria; SP, soluble protein; TP, total phenol; WSC, water-soluble carbohydrates; P, phosphorus; FAA, free amino acid; LL, leaf length; LW, leaf width; Ch, chlorophyll; MRC, moisture retention capacity; Tr, transpiration rate; Pn, net photosynthetic rate; Ci, intercellular carbon dioxide concentration; Gsw, pore conductivity of water vapor; Gtw, total conductivity of water vapor; Gtc, Total conductivity of carbon dioxide. IA1200, irrigation amount was 1,200 $m^3$ $hm^{-2}$; IA2400, irrigation amount was 2,400 $m^3$ $hm^{-2}$ . Asterisks indicate significant differences at $P < 0.05$ (*), $P < 0.01$ (**), and $P < 0.001$ (***), respectively.

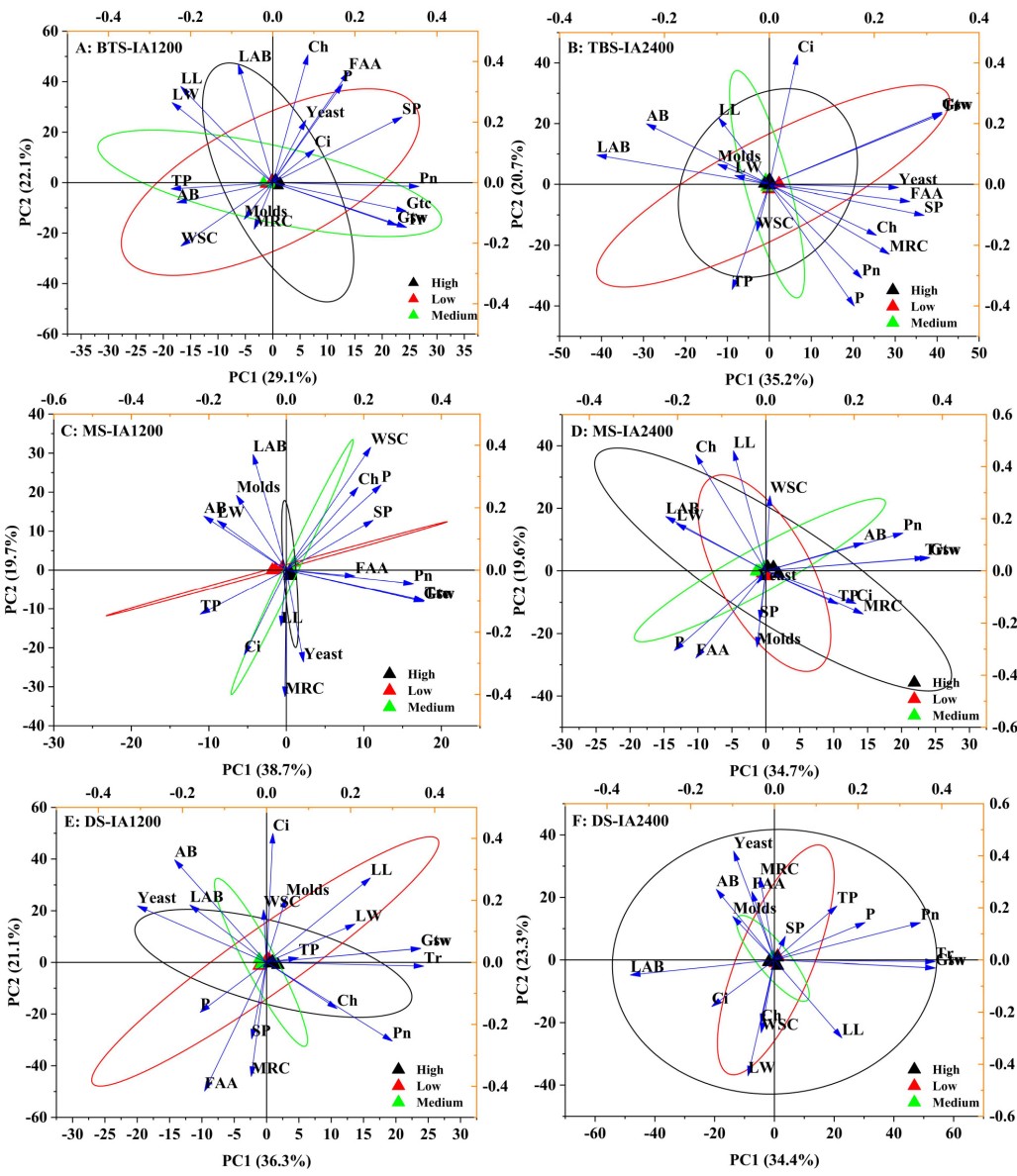

**Figure 2  Scatter diagram of principal component analysis of structure , microorganism compositions, chemical and physiological properties of different N application rate.** BTS-IA1200, irrigation amount was 1,200 m³ hm⁻² at the big trumpet stage; B: BTS-IA2400, irrigation amount was 2,400 m³ hm⁻² at the big trumpet stage; C: MS-IA1200, irrigation amount was 1,200 m³ hm⁻² at the milk stage; D: MS-IA2400, irrigation amount was 2,400 m³ hm⁻² at the milk stage; E: DS -IA1200, irrigation amount was 1,200 m³ hm⁻² at the dough stage; F: DS-IA2400, irrigation amount was 2,400 m³ hm⁻² at the dough stage. The ellipse indicates 95% confidence. AB, aerobic bacteria; LAB, lactic acid bacteria; SP, soluble protein; TP, total phenol; WSC, water- soluble carbohydrates; P, phosphorus; FAA, free amino acid; LL, leaf length; LW, leaf width; Ch, chlorophyll; MRC, moisture retention capacity; Tr, transpiration rate; Pn, net photosynthetic rate; Ci, intercellular carbon dioxide concentration; Gsw, pore conductivity of water vapor; Gtw, total conductivity of water vapor; Gtc, Total conductivity of carbon dioxide. Low, N application rate was 200 kg hm ⁻²; Medium, N application rate was 350 kg hm⁻²; High, N application rate was 500 kg hm⁻².

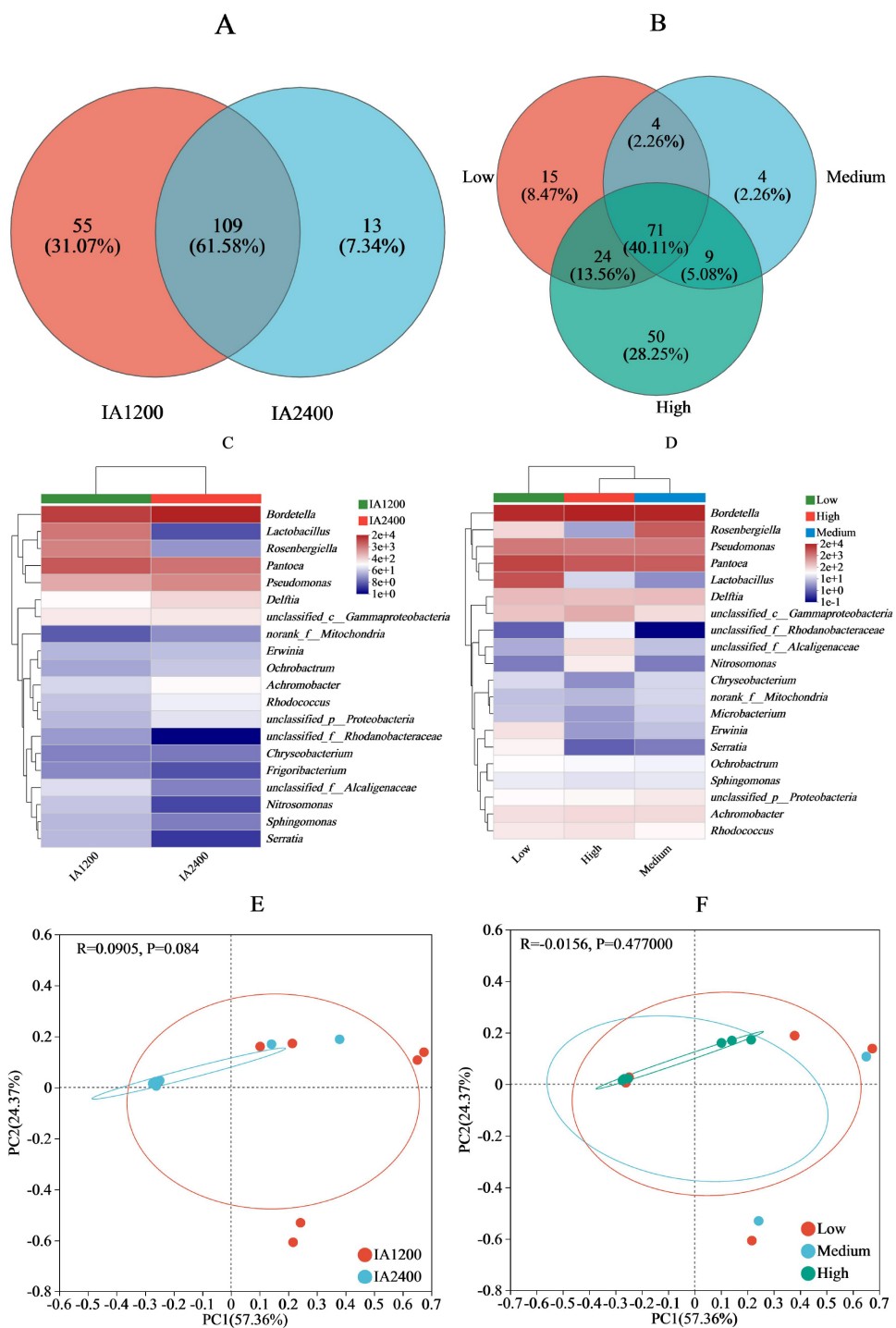

**Figure 3** **Effect of irrigation amount and N application rate on bacterial community (genus).** (A) and (B), different colors represent different irrigation amount or N application rate, overlapping numbers represent the number of species shared by multiple treatments, and non overlapping numbers represent the number of species unique to the corresponding treatment. (C) and (D), (continued on next page...)

(continued on next page...)

**Figure 3 (…continued)**
the $X$-axis indicates the irrigation amount or N application rate, the $y$-axis indicates the abundances of bacterial genera. (E) and (F), the first axis accounts for 57.36% of the total variance and the second for 24.37%. The original attributes, with their vectors intersecting at (0, 0), are also inserted. The length of each attribute vector is proportional to its contribution to the principal component axis. The ellipse indicates 95% confidence. IA1200, irrigation amount was 1,200 m³ hm⁻²; IA2400, irrigation amount was 2,400 m³ hm⁻²; Low, N application rate was 200 kg hm⁻²; Medium, N application rate was 350 kg hm⁻²; High, N application rate was 500 kg hm⁻².

fermentation quality. Among the many bacterial species with notable impacts, *Pantoea*, *Xanthomonas*, *Erwinia*, *Curtobacterium*, *Pseudomonas*, *Clavibacter*, *Enterobacter*, *Listeria*, *Citrobacter*, *Klebsiella*, *Bacillus*, and *Enterococcus* played crucial roles, each posing a challenge to the growth health of crops or the fermentation quality of silage feed in its unique way.

# DISCUSSION

## Compared with IA and NAR, the maturity stage has a greater impact on the survival and distribution of phyllosphere bacteria

At the dough stage, nutrients synthesized by photosynthesis in silage maize leaves are transferred to the grains, resulting in lower nutrient levels in the leaves (*Xu et al., 2010*). The water-soluble carbohydrate content at the dough stage was significantly higher compared with the big trumpet and milk stages. Even at the dough stage, the chlorophyll content in the leaves remained relatively high; therefore, despite the peak accumulation of dry matter in the grains, photosynthesis in the leaves continued, and the accumulated water-soluble carbohydrates were not transferred to the grains but accumulated in the leaves. Additionally, the dough stage experienced lower temperature, which also reduces the transfer rate of soluble carbohydrates from leaves to grains (*Pietrini et al., 1999*). In general, total phenols are important secondary metabolites in plants and possess antioxidant and antibacterial properties that enhance plant resistance (*Dehghanian et al., 2022*). In this study, the total phenol content in the leaves of silage maize was the highest at the dough stage. This can be explained by phenolic compounds' solubility and strong antioxidant properties, with a relatively low proportion of phenolic compounds used for polymerization during lignification. In addition, despite a decrease in the Tr at the dough stage, there was still a high oxygen demand; and it can protect cells from potential oxidative damage (*Mohamed, Lertrat & Suriharn, 2017*; *Randhir & Shetty, 2005*).

At the milk stage increased transpiration pull in maize leaves, further facilitating the transport of organic matter within the plant (*Ashraf et al., 2016*). Therefore, the Tr and Pn were highest at the milk stage (*Wu et al., 2023b*). These nutrients and physiological properties are the key to the survival and reproduction of leaf-associated microorganisms (*Tang et al., 2023*). In this study, aerobic bacteria were the most abundant, which is consistent with other findings (*Tang et al., 2022*; *Wu et al., 2023a*). Because the oxygen concentration on the leaf surface was relatively high, aerobic bacteria could utilize molecular oxygen as the final electron acceptor to carry out efficient energy-producing metabolic processes in aerobic environments. The milk stage had the highest number of aerobic bacteria, probably because of the strong metabolic activity at this growth stage (*Wu et*

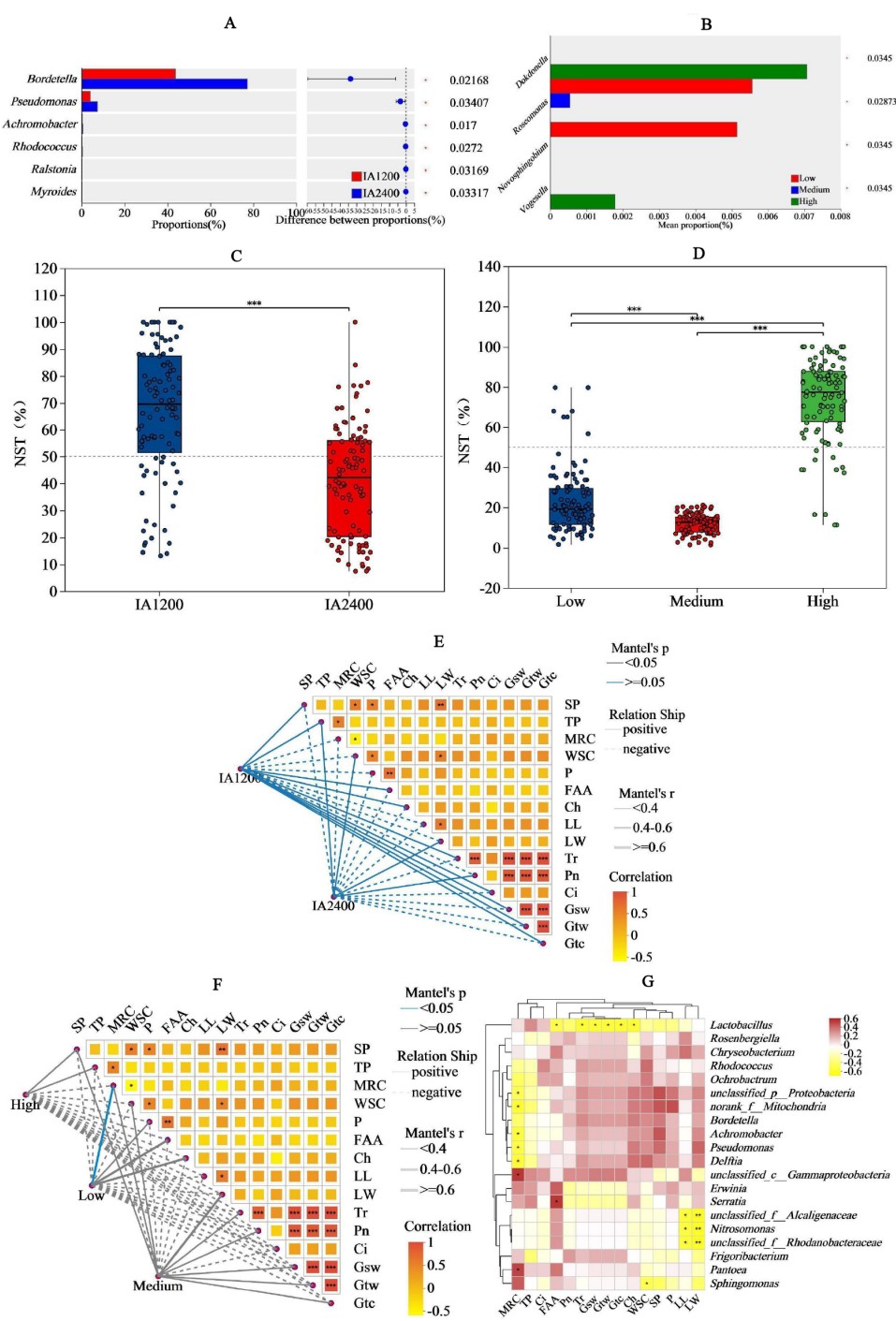

**Figure 4 Significance, normalized shuffle test index Mantel test and of bacteria relative abundance analyses of different irrigation amount and N application rate.** (A) and (B), *Y*-axis indicates the normalized shuffle test index, *X*-axis indicates the mean relative abundance 

**Figure 4 (…continued)**
in different subgroups of the species, and different coloured columns indicate different subgroups; *P*-values are shown on the rightmost side, and an asterisk (*) indicates $P < 0.05$. (C) and (D), $X$-axis indicates the irrigation amount or N application rate, $Y$-axis indicates the normalized shuffle test index, the dotted lines were thresholds for deterministic and random divisions, demonstrating significance markers for between-group analysis of variance. NST, normalized shuffle test index. (E) and (F), the lines represent the correlation among bacterial communities and environmental factors, while the heatmap represents the correlation among environmental factors. In the heatmap, different colors represent positive or negative correlations, and color depth represents the magnitude of positive or negative correlations. * $P < 0.05$, ** $P < 0.01$, *** $P < 0.001$. G, SP, soluble protein; TP, total phenol; WSC, water- soluble carbohydrates; P, phosphorus; FAA, free amino acid; Ch, chlorophyll; LL, leaf length; LW, leaf width; Ch, chlorophyll; MRC, moisture retention capacity; Tr, t ranspiration rate; Pn, net photosynthetic rate; Ci, i ntercellular carbon dioxide concentration; Gsw, pore conductivity of water vapor; Gtw, total conductivity of water vapor; Gtc, total conductivity of carbon dioxide. IA1200, irrigation amount was 1,200 m³ hm⁻²; IA2400, irrigation amount was 2,400 m³ hm⁻²; Low, N application rate was 200 kg hm⁻²; Medium, N application rate was 350 kg hm⁻²; High, N application rate was 500 kg hm⁻².

**Table 4    Effects of environmental factors on the relative abundance of bacteria.**

| Items | Fitted equation | R² | P |
|---|---|---|---|
| Soluble protein | $y = 1,095.771x-7,509.628$ | 0.0815 | 0.2509 |
| Total phenol | $y = -1,571.971x+6,444.471$ | 0.0188 | 0.5878 |
| Water- soluble carbohydrates | $y = 744.946x-5,835.110$ | 0.0788 | 0.2593 |
| Phosphorus | $y = 12.889x-4,065.097$ | 0.0135 | 0.6464 |
| Free amino acid | $y = -501.540x+4,772.771$ | 0.0402 | 0.425 |
| Chlorophyll | $y = 3,855.730x-9,597.608$ | 0.0360 | 0.4,508 |
| Leaf length | $y = -28.535x+2,805.895$ | 0.0011 | 0.8948 |
| Leaf width | $y = 1,344.214x-15,122.408$ | 0.0516 | 0.3647 |
| Moisture retention capacity | $y = -690.750x+18,883.534$ | 0.2386 | 0.0397 |
| Transpiration rate | $y = 1980.857x-6,846.554$ | 0.0437 | 0.4054 |
| Net photosynthetic rate | $y = 277.465x-8,668.258$ | 0.0147 | 0.6314 |
| Intercellular carbon dioxide concentration | $y = -47.700x+3,773.37$ | 0.0071 | 0.7394 |
| Pore conductivity of water vapor | $y = 31.858x-5,174.955$ | 0.0257 | 0.5249 |
| Total conductivity of water vapor | $y = 33.217x-5,206.777$ | 0.0252 | 0.5293 |
| Total conductivity of carbon dioxide | $y = 52.836x-5,202.323$ | 0.0253 | 0.5287 |

*al., 2023a*), which enhanced the nutrient supply capacity for aerobic bacteria. A higher Tr improves the moist environment on the leaf surface, facilitating the reproduction and survival of the bacteria. Furthermore, the big trumpet and milk stages occur during the warm and humid seasons, which is conducive to microbial survival (*Guan et al., 2018*). By contrast, the dough stage is often associated with lower temperatures and humidity, leading to fewer bacteria in the phyllosphere of silage maize. However, the dough stage exhibited the highest number of molds. It is possible that some molds are more active in drier environments (*Wu & Wong, 2022*).

Xu et al. (2025), *PeerJ*, DOI 10.7717/peerj.19663

**Table 5** Common harmful bacteria in fresh materials and natural silage and its on silage fermentation characteristics of forage on meta-analysis.

| Material type | Species name | Function | Meta-regression parameters on lactic acid concentration in silage | Meta-regression parameters on ammonia-N concentration in silage | Meta-regression parameters on butyric acid concentration in silage | Note |
|---|---|---|---|---|---|---|
| | *Pantoea* | Maize white spot disease (*Xing et al., 2023*); producing acetic acid using sugar and amino acids during the fermentation process (*Jiang et al., 2023*). | $P < 0.01$, $R^2 = 0.2713$, SEM = 0.15 | $P < 0.01$, $R^2 = 0.4718$, SEM = 0.27 | $P < 0.05$, $R^2 = 0.2544$, SEM = 0.09 | The relative abundance of this species ranked fourth in this study. |
| | *Xanthomonas* | Foliar diseases (*Ruiz et al., 2023*); showed a negative correlation with the production of lactic acid and acetic acid in silage (*Franco et al., 2022b*). | $P < 0.05$, $R^2 = 0.4895$, SEM = 3.55 | $P > 0.05$, $R^2 = 0.0043$, SEM = 1.60 | $P > 0.05$, $R^2 = 0.018$, SEM = 0.22 | – |
| | *Erwinia* | Bacterial stalk rot (*Shahid et al., 2024*); showed a negative correlation with the production of ammonia N, lactic acid and acetic acid in silage (*Franco et al., 2022a*). | $P < 0.05$, $R^2 = 0.4724$, SEM = 0.93 | $P > 0.05$, $R^2 = 0.0021$, SEM = 2.20 | $P > 0.05$, $R^2 = 0.0411$, SEM = 0.29 | The relative abundance of this species ranked fourteenth in this study. |
| | *Curtobacterium* | Tar spot disease (*Johnson et al., 2023*); decompose cellulose and hemicellulose in silage materials (*Xie et al., 2022*). | $P < 0.05$, $R^2 = 0.2870$, SEM = 0.36 | $P > 0.05$, $R^2 = 0.013$, SEM = 0.14 | $P < 0.05$, $R^2 = 0.167$, SEM = 0.54 | – |
| | *Pseudomonas* | Leaf bligh (*Lorenzetti et al., 2020*); cause the corruption and mildew of silage (*Jiang et al., 2020*). | $P < 0.05$, $R^2 = 0.5578$, SEM = 0.39 | $P < 0.001$, $R^2 = 0.5121$, SEM = 0.16 | $P < 0.001$, $R^2 = 0.3411$, SEM = 0.16 | The relative abundance of this species ranked seventeenth in this study. |
| | *Clavibacter* | Wilt and leaf bligh (*Bauske & Friskop, 2021*); the negative impact on the quality of feed and animal health (*Kennang Ouamba et al., 2022*). | $P < 0.05$, $R^2 = 0.4119$, SEM = 0.28 | $P < 0.05$, $R^2 = 0.2411$, SEM = 0.65 | $P < 0.01$, $R^2 = 0.1611$, SEM = 0.17 | – |
| Fresh sample | *Enterobacter* | Capable of hydrocarbon degradation and efficiently colonized the rhizosphere and plant interior (*Yousaf et al., 2011*); compete with LAB for available substrates during ensiling and reduce feed quality (*Guo et al., 2023*). | $P < 0.01$, $R^2 = 0.274$, SEM = 0.57 | $P < 0.01$, $R^2 = 0.3119$, SEM = 1.18 | $P < 0.01$, $R^2 = 0.2478$, SEM = 0.13 | The relative abundance ranked after the top 20 in this study. |
| | *Listeria* | The leading cause of bacterial-linked foodborne mortality (*Truong et al., 2021*). | – | – | – | – |
| | *Citrobacter* | Improved plant shoot length, root length, fresh weight, dry weight, and chlorophyll a content (14%–24%) and chlorophyll b (*Ajmal et al., 2022*). | – | – | – | – |
| | *Klebsiella* | Caused typical top rot symptoms (*Huang et al., 2016*). | – | – | – | – |

Peer J

**Table 5** (*continued*)

| Material type | Species name | Function | Meta-regression parameters on lactic acid concentration in silage | Meta-regression parameters on ammonia-N concentration in silage | Meta-regression parameters on butyric acid concentration in silage | Note |
|---|---|---|---|---|---|---|
| | *Enterobacter* | Compete with the lactic acid bacteria for nutrients (*Wang et al., 2019*). | $P > 0.05$, $R^2 = 0.0669$, SEM = 2.66 | $P > 0.05$, $R^2 = 0.043$, SEM = 4.95 | $P > 0.05$, $R^2 = 0.022$, SEM = 1.27 | – |
| | *Bacillus and Enterococcus* | Protein-degrading bacteria (*Hao et al., 2020*). | $P < 0.05$, $R^2 = 0.2745$, SEM = 0.95 | $P < 0.05$, $R^2 = 0.1544$, SEM = 1.05 | $P > 0.05$, $R^2 = 0.0028$, SEM = 0.01 | – |
| | *Pseudomonas* | Biogenic amine production (*Li et al., 2024*). | $P < 0.001$, $R^2 = 0.2124$, SEM = 2.43 | $P < 0.001$, $R^2 = 0.3512$, SEM = 4.29 | $P > 0.05$, $R^2 = 0.011$, SEM = 0.11 | – |
| | *Clostridium* | Efflux of a large amount of $NH_4^+$ (*Zhou et al., 2024*). | $P < 0.05$, $R^2 = 0.2712$, SEM = 0.73 | $P < 0.05$, $R^2 = 0.1718$, SEM = 2.55 | $P < 0.05$, $R^2 = 0.1522$, SEM = 0.24 | – |
| | *Listeria* | A foodborne pathogen, an agent that causes listeriosis: a serious invasive disease that affects both humans and a wide range of animals (*Nucera et al., 2016*). | $P > 0.05$, $R^2 = 0.0034$, SEM = 0.27 | $P > 0.05$, $R^2 = 0.013$, SEM = 3.45 | $P > 0.05$, $R^2 = 0.007$, SEM = 4.58 | – |
| Natural silage without additives treatment | *Citrobacter* | Can produce protease to degrade protein and decarboxylate amino acids (*Zhang et al., 2019a*). | $P < 0.001$, $R^2 = 0.1224$, SEM = 0.72 | $P < 0.001$, $R^2 = 0.2387$, SEM = 0.05 | $P < 0.05$, $R^2 = 0.1483$, SEM = 0.26 | – |
| | *Klebsiella* | Can reduce $NO_2^-$ to $N_2O$ and $NH_4^+$ under anaerobic conditions (*Hu et al., 2024*). | $P < 0.01$, $R^2 = 0.2271$, SEM = 0.11 | $P < 0.01$, $R^2 = 0.2213$, SEM = 0.16 | $P > 0.05$, $R^2 = 0.0071$, SEM = 0.04 | – |

**Notes.**

SEM, standard error of the means.

## Low IA was beneficial for the survival and reproduction of plant pathogens

Photosynthesis is the primary pathway through which plants produce organic substances, including proteins (*Ávila et al., 2023*; *Chai et al., 2015*), and moderate irrigation can provide sufficient water and nutrients to plants, thereby facilitating their growth and nutrient synthesis (*Waraich et al., 2011*). In the present study, neither IA1,200 nor IA2,400 imposed stress on silage maize, and normal growth was maintained under both irrigation levels. This was confirmed by the subtle difference in total phenolic content between the two irrigation levels. In general, under drought conditions, plants must synthesize phenolic compounds to cope with the physiological stress caused by drought and reduce transpiration. In contrast, an adequate water supply leads to the upregulation of genes encoding enzymes involved in brassinosteroid synthesis, thereby suppressing the synthesis of total phenols (*Barreales et al., 2019*). When available water decreases, plants reduce their stomatal density to minimize water loss (*Bertolino, Caine & Gray, 2019*), which is unfavorable for forming a humid leaf environment necessary for microbial survival. Bacteria facing drought stress can enhance their stress tolerance by increasing the thickness of their cell walls. Therefore, in more arid environments, a small number of bacteria, such as lactic acid bacteria, can still be detected in the phyllosphere of forage plants (*Bao et al., 2023*). Furthermore, differences in nutrition and physiological properties at different growth stages exacerbate the impact of IA on the number of lactic acid bacteria ($P < 0.05$) (Table 3). Additionally, some bacteria possess the ability to produce extracellular polysaccharides, and the synthesis of these polysaccharides enhances their stress tolerance (*Nwodo, Green & Okoh, 2012*) and regulates the expression of drought stress-related genes (*Daranas et al., 2018*), which is a crucial mechanism for survival in drought environments. In the present study, a higher number of aerobic and lactic acid bacteria was detected in IA1,200 compared to IA2,400, suggesting that the environment of IA1,200 did not face drought stress or that the bacteria in its plants possessed greater stress tolerance and survival ability. However, as the amount of irrigation increased from IA1,200 to IA2,400, the numbers of aerobic and lactic acid bacteria decreased; indicating that excessive irrigation was unfavorable for their growth. High irrigation levels reduce the oxygen content of plants, which is disadvantageous for the survival of aerobic bacteria (*Irmak, 2008*). Some studies have also confirmed that the microbial communities attached to silage maize are mainly composed of aerobic bacteria, such as *Paenibacillus*, *Sphingomonas*, *Rhizobiaceae*, and *Buchnera* (*Kadivar & Stapleton, 2003*).

Irrigation levels also have the ability to regulate the structure of leaves (*Zheng et al., 2010*), which can indirectly affect a series of ecological characteristics on the leaf surface, such as temperature (*Lindow & Brandl, 2003*), hydrophilicity, and the type and quantity of nutrients (*Bickford, 2016*). These characteristics play crucial roles in bacteria's survival and reproductive environment. Notably, IA1,200 had a higher ($P < 0.001$) NST index than IA2,400 which was closely related to the higher OTU levels in IA1,200. This result reflects a more intense competitive relationship among bacteria and higher environmental complexity in the IA1,200 environment. Despite the predominantly positive ($P < 0.05$) correlation between IA1,200s impact on environmental factors, improving environmental

conditions intensified competition among bacteria for living space and nutrients. In contrast, the effect of IA2,400 on environmental factors was often negative, which enhances the stability of the bacterial network structure (*Fan et al., 2018*). In this study, there was a significantly higher ($P < 0.05$) relative abundance of beneficial (*Pseudomonas* (*Mehnaz & Lazarovits, 2006*), *Rhodococcus* (*Cserháti et al., 2013*), *Achromobacter* (*Verhage, 2020*), and *Myroides* (*Kaur & Kaur, 2021*)) in IA2,400 compared with in IA1,200. *Erwinia* (*Piqué et al., 2015*) and *Serratia* (*Shikov et al., 2023*) have both been considered pathogens that cause plant diseases, and they had the highest relative abundance in the IA1,200. Therefore, the results indicate that low IA was beneficial for the survival and reproduction of plant pathogens, and this might increase the risk of reducing silage's nutrition and hygienic quality. Furthermore, the results of the meta-analysis profoundly reveal that these harmful bacteria not only jeopardize the safety of forage yield and nutritional structure but also pose a severe challenge to the fermentation quality of silage (Table 5). Among the top 20 bacterial species in terms of relative abundance identified in this study, we discovered three crucial problematic organisms that have significantly adverse effects on the fermentation quality of silage (*Franco et al., 2022a*; *Jiang et al., 2020*; *Jiang et al., 2023*). This discovery further underscores that irrigation and nitrogen fertilizer management, in addition to influencing yield and nutritional composition, also exert an indirect and significant impact on the fermentation quality of silage through the modulation of microbial communities.

## Low and high NARs were beneficial for plant pathogens' survival and reproduction

Nitrogen improves chlorophyll and enzyme activity in leaves (*Xing et al., 2023*), enhancing photosynthesis and crop yield (*Ruiz et al., 2023*). However, in our study, the NAR's impact on silage maize leaf physiology and chemistry was minimal (Tables 1 and 2), suggesting it wasn't the main limiting factor for nutrient synthesis. Although photosynthetic parameters increased with NAR (*Xing et al., 2023*), the leaf surface, despite its volatility, supports bacteria closely linked to plant growth, many of which rely on the *cydABCD* operon for aerobic respiration (*Shahid et al., 2024*). This explains the predominance of aerobic bacteria in the phyllosphere. The NAR's effect on phyllosphere bacteria was insignificant, with lactic acid bacteria numbers typically low, differing from some previous findings (*Johnson et al., 2023*). Potential reasons include crop type (*Bauske & Friskop, 2021*; *Lorenzetti et al., 2020*; *Zhang et al., 2019b*). N demand, and the saturation point beyond which additional N does not increase bacterial numbers but may decrease diversity (*Chen, Dong & Zhang, 2021*). This trend wasn't directly observed in our study but could be reflected in other ways, such as the highest OTU count under high NAR, indicating increased bacterial diversity without a significant rise in total numbers. The high NAR was negatively correlated with over 86.7% of environmental factors, suggesting intensified bacterial competition and network instability, confirmed by the highest NST index (Fig. 4). High NAR also saw the highest relative abundance of potentially harmful microbes like *Dokdonella* and *Vogesella* (*Hao et al., 2020*) ($P < 0.05$), while plant pathogens like *Erwinia* and *Serratia* were more prevalent at low NAR. This implies that both low and high NARs could favor pathogen survival, risking silage hygiene and animal health. Bacterial competition likely stems from limited

resources, and when these are scarce or excessive, competition intensifies, potentially leading to the dominance of some populations and the decline of others. This imbalance could disrupt phyllospheric ecosystem stability and its resilience to external stressors. In conclusion, while the NAR's impact on microbial numbers was not significant, it may affect the stability of the bacterial network structure. In addition, there were still uncertainties regarding nitrogen loss and the assessment of its availability in this study. Future research should focus on fertilization strategies that enhance nitrogen use efficiency and further investigate their impact on the structure of microbial communities.

### Based on yield and plant phytosanitation considerations, IA1,200 and medium NAR were the optimal combination

Forage plants frequently face issues related to diseases caused by bacteria or concerns about their cleanliness during growth and silage production, resulting in substantial financial losses for growers (Table 5). In this research, various bacteria were recognized as impacting the nutritional and cleanliness of crops or forage, encompassing *Ralstonia*, (*Ahmed et al., 2022*) *Dokdonella*, *Vogesella* (*Xing et al., 2023*), *Erwinia* (*Piqué et al., 2015*), *Serratia* (*Shikov et al., 2023*), and *Bordetella* (*Ruiz et al., 2023*). Among these identified species, the prevalence of all except *Bordetella* was found to be minimal in the study (Figs. 3 and 4). Interestingly, a significant positive correlation was only observed between the content of free amino acids and the prevalence of *Serratia* (Fig. 4). This suggests that the environmental variables influenced by the IA and NAR in this study did not generally favor the growth of harmful bacteria, potentially due to the constrained variations in IA and NAR. It is essential for growers to take into account yield-related factors. The study identified the most effective combination as an IA of 2,400 $m^3$ $hm^{-2}$ and a NAR of 240 kg $hm^{-2}$ for maximizing yield (Table S1). Moving forward, research should delve into the interplay between irrigation practices, NAR, and the functions of phyllosphere microorganisms, which include both bacteria and fungi, to gain a deeper understanding of the phyllosphere ecosystem's functionality and stability. Regrettably, this study did not extend its scope to examine the diversity and roles of fungi in plant health and their interactions with plants under varying IA and NAR conditions. Further studies are necessary to investigate the potential ecological impacts and management strategies related to the enhancement of beneficial microorganisms or the reduction of detrimental ones.

## CONCLUSION

The maturity stage significantly influences phyllosphere bacteria more than irrigation or nitrogen application rates. Aerobic bacteria dominate, especially at the milk stage, due to higher metabolic activity and transpiration. Low irrigation favors pathogen survival, while high irrigation reduces oxygen, detrimental to aerobic bacteria. Both low and high nitrogen application rates may benefit pathogens, impacting silage hygiene. Optimal conditions for yield and phytosanitation are moderate irrigation and medium nitrogen application rates. Moreover, the predictive analysis from meta-analysis reveals that the influence of irrigation and nitrogen management on the fermentation quality of silage is not merely limited to affecting forage nutrition; more crucially, they potential impact the fermentation quality

of silage by finely regulating the bacterial species and their relative abundances. Overall, the findings of the meta-analysis were validated and supported by the results of this study, further strengthening reliability and practicalness.

Based on the yield, plant health, and potential fermentation quality of silage maize, combined with verification from a meta-analysis, this study recommend using an irrigation amount of 2,400 $m^3$ $hm^{-2}$ and an NAR of 240 kg $hm^{-2}$ for silage maize cultivation, with harvest at the milk stage. This recommendation aimed to optimize the agricultural performance of silage maize while improving fermentation quality of the silage, ensuring feed safety and high nutritional quality.

### Abbreviations

| | |
|---|---|
| **N** | nitrogen |
| **NST** | normalized shuffle test |
| **OUT** | operational taxonomic unit |
| **PCR** | polymerase chain reaction |
| **PCA** | principal component analysis. |

### Funding

This work was supported by the Yunnan Revitalization Talents Support Plan (XDYC-CYCX-2022-0036), Yunnan Young Talents (2023), and Eryuan County Forage Industry Science and Technology Mission (202304BI090008). The funders had no role in study design, data collection and analysis, decision to publish, or preparation of the manuscript.

### Grant Disclosures

The following grant information was disclosed by the authors:
Yunnan Revitalization Talents Support Plan (2022).
Yunnan Young Talents (2023).
Eryuan County Forage Industry Science and Technology Mission: 202304BI090008.

### Competing Interests

The authors declare there are no competing interests.

### Author Contributions

- Liuxing Xu conceived and designed the experiments, prepared figures and/or tables, and approved the final draft.
- Changjing Chen performed the experiments, prepared figures and/or tables, and approved the final draft.
- Chenggang He analyzed the data, prepared figures and/or tables, and approved the final draft.
- Ahmed M. Abd El Tawab performed the experiments, prepared figures and/or tables, and approved the final draft.

- Qinhua Liu performed the experiments, prepared figures and/or tables, authored or reviewed drafts of the article, and approved the final draft.
- Hua Jiang performed the experiments, prepared figures and/or tables, and approved the final draft.

## DNA Deposition

The following information was supplied regarding the deposition of DNA sequences:
The data is available in the National Microbiology Data Center (NMDC): NMDC40056008 to NMDC40056025.

## Data Availability

The experimental data are available in the Supplementary File.

## Supplemental Information

Supplemental information for this article can be found online at http://dx.doi.org/10.7717/peerj.19663#supplemental-information.

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
