# Peer review of "Effects of irrigation and nitrogen management on phyllosphere microbial communities of silage maize"

_PeerJ, doi:10.7717/peerj.19663_

## Round 0.1 · original submission · Major Revisions

Dear Dr. Xu, I kindly ask you to carefully correct and supplement the manuscript in accordance with the reviewers' comments. I hope that this will give them the opportunity to approve this manuscript for publication.

**Language Note:** The review process has identified that the English language must be improved. PeerJ can provide language editing services - please contact us at [email protected] for pricing (be sure to provide your manuscript number and title). Alternatively, you should make your own arrangements to improve the language quality and provide details in your response letter. – PeerJ Staff

·

Basic reporting

Title of the article: ‘influence on properties’ is logically erroneous, because a property means the presence or absence of a certain quality in an object that distinguishes it from others. The influence on the quantitative manifestation of a property (and not on the property as such) is unique and is not formally the subject of scientific investigation. The scientific sense has an impact on a phenomenon or a process (process dynamics)

Abbreviations are not acceptable in the abstract.
Excessive detail (P<0.05) should also be avoided in the abstract

«Irrigation is another key factor that affects plant growth» – this sentence is not quite correct; irrigation can provide optimal water conditions for achieving the desired crop yield, but does not directly affect plant growth
A similar comment on the phrase: ‘irrigation influence the bacterial communities’

The phrase also breaks the logic: «Specifically, it explores how these agronomic practices shape the composition and function of phyllosphere bacteria, and the subsequent effects on silage fermentation quality, as well as its nutritional and hygienic quality» – agronomic practices are designed to influence economically important indicators of agricultural activity: i.e. fertilisers and irrigation are likely to affect the quality of agricultural products. And to explain this impact, we should consider the hypothesis that the dynamics of phyllosphere bacteria communities can explain the mechanism of such an impact.

Experimental design

Lines 96-97 - abbreviations should be given immediately with the full name of the term
What is the name of the soil according to WRB?

How many foam boxes were tested in the experiment?

«Leaves were collected at the big trumpet (August 13, 2022), milk (September 17, 2022), and dough (November 13, 2022) stages» – How did the collection of leaves affect the physiological state of the plants?

Validity of the findings

The authors' use of statistical procedures is not logical. They indicate that they can perform certain statistical procedures on the data matrices, but the article does not explain what the point of such procedures is. The absence of information on the number of repeats makes these results extremely questionable

Table 1. Chemical properties, leaf length, and leaf width of different maturity stage, irrigation amount, and N application rate – the number of replications is not specified and what statistics are marked with "±"

What is the point of using principal component analysis? Are the first two principal components statistically significant? What is the number of statistically significant principal components? How do several different principal component analyses for different experimental designs explain the observed mechanisms?

Figure 3. Effect of irrigation amount and N application rate on bacterial community (genus) – The authors present a certain procedure for assessing the impact of environmental factors on microbial communities. It is necessary to either provide literature references to the authors who first proposed such a procedure or provide your own vision of the procedure, if it is proposed for the first time. In general, there are other procedures for achieving this task, which should be used unless the authors have a valid reason for doing so that should be explained

Additional comments

The article gives the impression of a study whose results are too cumbersome to present. The authors should not overcomplicate the presentation of the results if it is not justified. The procedures used mostly violate the conditions under which their application gives relevant and reasonable results (number of replications, normality of the sample, selection of the best statistical methods for solving the tasks: a very large number of calculations can be performed with the data set, but the authors' responsibility is to propose the best approach and justify it)

Reviewer 2 ·

Basic reporting

The authors have done a lot of work, but the results are not obvious. In fact, it is a model experiment, the results of which are difficult to transfer to production conditions. Moreover, the discussion does not address possible inaccuracies in the results.
The main problem with corn is fungal diseases, which also cause problems with wheat and barley in rotations. Corn monoculture increases the complexity of controlling these diseases and producing high quality silage. The authors' work does not provide solutions to these problems.
In the technologies for obtaining high quality silage, the main indicators include the accumulation of dry matter, but this is not mentioned in the article.

Experimental design

The conclusions of the article are based on 2 plants per pot per replication of each variant.

Validity of the findings

see paragraph 2

Additional comments

Surface application of urea is not a good solution, there are large losses due to photodegradation and urease activity, and it is difficult to determine the appropriate content of nitrogen forms in the soil.
It is necessary to specify the BBCH phases in the text.
Specify the time of sampling, whether there is dew in the morning - it is important for reproducing the results.
LI-6800 - specify the detailed name of the device, manufacturer.

---

## Round 0.2 · Major Revisions

Dear Dr. Xu, I ask you to make corrections to the manuscript in accordance with the reviewer's comments (both in the first and second reviews). I hope that the next version of your article will already be accepted for publication.

·

Basic reporting

The authors have implemented all the recommendations of the reviewer. The quality of the manuscript has been significantly improved.

Experimental design

The authors have implemented all the recommendations of the reviewer. The quality of the manuscript has been significantly improved.

Validity of the findings

The authors have implemented all the recommendations of the reviewer. The quality of the manuscript has been significantly improved.

Additional comments

The authors have implemented all the recommendations of the reviewer. The quality of the manuscript has been significantly improved. I recommend the article for publication

Reviewer 2 ·

Basic reporting

The authors have made a large number of improvements that are worth agreeing with.
BBCH - the first mention is full, the next one is abbreviated.

Chlorophyll: Standard error of the means - 2.70? Is that a mistake?

Tables should contain all necessary explanations. In some tables, the authors did not provide the necessary statistical calculations. For example, for chlorophyll: the top 3 rows have calculations, the bottom rows do not. Do the authors lack primary data for the calculations? Below: 2.08 +/- an error.

Experimental design

no comment

Validity of the findings

See N1

Additional comments

no comment

---

## Round 0.3 · accepted · Accept

Dear Dr. Xu, I am pleased to inform you that this article has been accepted for publication.

Reviewer 2 ·

Basic reporting

Captions to figures and tables in the pdf contain numerous unnecessary spaces.
Please check line 618.

Experimental design

no comment

Validity of the findings

no comment